# Unsupervised Part Discovery from Contrastive Reconstruction

**Subhabrata Choudhury**      **Iro Laina**      **Christian Rupprecht**      **Andrea Vedaldi**

Visual Geometry Group
University of Oxford
Oxford, UK
`subha,iro,chrisr,vedaldi@robots.ox.ac.uk`

## Abstract

The goal of self-supervised visual representation learning is to learn strong, transferable image representations, with the majority of research focusing on object or scene level. On the other hand, representation learning at part level has received significantly less attention. In this paper, we propose an unsupervised approach to object part discovery and segmentation and make three contributions. First, we construct a proxy task through a set of objectives that encourages the model to learn a meaningful decomposition of the image into its parts. Secondly, prior work argues for reconstructing or clustering pre-computed features as a proxy to parts; we show empirically that this alone is unlikely to find meaningful parts; mainly because of their low resolution and the tendency of classification networks to spatially smear out information. We suggest that image reconstruction at the level of pixels can alleviate this problem, acting as a complementary cue. Lastly, we show that the standard evaluation based on keypoint regression does not correlate well with segmentation quality and thus introduce different metrics, NMI and ARI, that better characterize the decomposition of objects into parts. Our method yields semantic parts which are consistent across fine-grained but visually distinct categories, outperforming the state of the art on three benchmark datasets. Code is available at the project page: https://www.robots.ox.ac.uk/~vgg/research/unsup-parts/.

## 1   Introduction

Humans perceive the world as a collection of distinct objects. When we interact with an object, we naturally perceive the different parts it consists of. In visual scene understanding, parts provide intermediate representations that are more invariant to changes in object pose, orientation, camera view or lighting than the objects themselves. They are useful in analyzing objects for higher level tasks, such as fine-grained recognition, manipulation etc. However, supervised learning of parts requires manual annotations, which are slow, expensive and infeasible to collect for the almost unlimited variety of objects in the real world. Thus, unsupervised part discovery and segmentation has recently gained the interest of the community. We thus consider the problem of automatically discovering the parts of visual object classes: given a collection of images of a certain object category (e.g., birds) and corresponding object masks, we want to learn to decompose an object into a collection of repeatable and informative parts.

It is important to define and understand the nature of parts before we begin describing approaches to part discovery. While there is no universally accepted formal definition for what constitutes a "part", the nature of objects and object parts is accepted as different. For example, for Gibson [26], an object is "detachable", *i.e.* something that, at least conceptually, can be picked up and moved to a different place irrespective of the rest of the scene. Parts, in contrast, are constituent elements of an object, and

35th Conference on Neural Information Processing Systems (NeurIPS 2021).

cannot be removed without breaking the object, *i.e.* they are essential to the object and occur across most instances of the same object category.

Unsupervised part discovery requires suitable inductive principles and the choice of these principles defines the nature of the parts that will be discovered. Parts could, for example, be defined based on motion following the principle of common fate in Gestalt psychology (*i.e.* what moves together belongs together) [66, 73], or they could be defined based on visual appearance or function. Here, we are interested in *semantic* parts across different instances of an object category (e.g., birds, cars, etc.) and combine three simple learning principles as "part proxy": (a) consistency to transformation (equivariance), (b) visual consistency (or self-similarity), and (c) distinctiveness among different parts.

Prior work has suggested that useful cues for part discovery can be obtained from pre-trained neural networks [2, 28]. These networks can in fact be used as a dense feature extractors and the feature responses can be clustered or otherwise decomposed to identify parts [14, 37]. In particular, [37] learn part prototypes, and make the latter orthogonal to avoid parts collapsing into a single one.

In this paper, we revisit and improve such concepts. We make the following contributions. First, we show that contrastive learning can be employed as an effective tool to decompose objects into diverse and yet consistent parts. In particular, we seek parts whose feature responses are *homogeneous* within the same or different occurrences of the *same part type*, while at the same time being *distinctive* for *different types* of parts. A second contribution is to discuss whether clustering pre-trained features is indeed sufficient for part discovery. To this end, we show that simply clustering dense features sometimes captures obviously self-similar structures, such as image edges, rather than meaningful parts (Section 3.2). This is somewhat intrinsic to using pre-trained feed-forward local features, as these can only analyze a fixed image neighborhood and thus pick up the pattern which is most obvious within their aperture. As a complementary cue, we thus suggest to look at the visual consistency of parts. The idea is that most parts are visually homogeneous, sharing a color or texture. A generative model of the part appearance may thus be able to detect part membership at the level of individual pixels. We show, in particular, that even very simple models that assume color consistency are complementary and beneficial when added to feature-based grouping.

Finally, we consider the problem of *assessing* automated part discovery. An issue is the relative scarcity of data labelled with part segmentation. Another one, technically more challenging, is the fact that parts that are discovered without supervision may not necessarily correspond to the parts that a human annotator would assign to an image. This makes the use of manual part annotations for evaluation tricky. Prior work in the area has thus assessed the discovered parts via proxy tasks, such as learning keypoint predictors, using supervision. The idea is that, if parts are consistent, they should be good predictors of other geometric primitives. Unfortunately, as we show empirically, transferring parts to keypoints is unlikely to provide a meaningful metric for the quality of the part segments. We show, for instance, that knowledge of a *single* keypoint provides a better predictor of other keypoints than *any* of the previous unsupervised models.

To address this issue, we propose a new evaluation protocol. We still use keypoints as they are readily available, or ground-truth part segmentation when possible; however, instead of learning to regress such ground truth annotations, we simply measure the co-occurrence statistics of the predicted parts and these annotations using Normalized Mutual Information and Adjusted Rand Index. The latter require the learned parts to be geometrically consistent and distinctive regardless of whether they are in one-to-one correspondence with manually provided labels and results in a more meaningful measure for this task.

Empirically, we demonstrate that these improvements lead to stronger automated part discovery than prior work on standard benchmarks.

## 2   Related work

There exists a vast amount of literature that studies the problem of decomposing a scene into objects and objects into parts, with or without supervision. Next, we discuss these lines of work with a focus on unsupervised approaches.

**Unsupervised scene decomposition.**   Unsupervised scene decomposition methods aim to spatially decompose a scene into a variable number of components (segments), e.g., individual objects and

background. This is typically achieved by encoding scenes into object-centric representations which are then decoded against a reconstruction objective, often in a variational framework [44]. Representative methods leverage sequential attention [5, 20], iterative inference [19, 29, 55], spatial attention [16, 52] and physical properties [3], or extend towards temporal sequences [3, 17, 40, 41, 46]. Discriminative approaches [45] also exist, using an object-level contrastive objective. It has been shown that, currently, most of these models perform well on simple, synthetic scenes but struggle with higher visual complexity [43].

There are key conceptual differences between decomposing a scene into objects and decomposing an object into its parts. First, the objects of a scene typically appear in an arbitrary arrangement, whereas the presence and the layout of object parts is generally constrained (e.g., the arms of a human connect to the torso). As such, parts are constituent elements of an object and they occur consistently across most instances of an object category. Moreover, the segments obtained by the systems described above are orderless and do not have a "type" assigned to them, in contrast to parts which usually refer to specific, nameable entities (e.g., head vs. beak). These differences lead to sufficiently different statistics and technical constraints, which make it difficult to envision methods that can do well both at scene-level and object-level.

**Part discovery and segmentation.** Prior to deep learning, part-based models [15, 21–23] played a major role in problems such as object detection and recognition. In the deep learning era, part discovery remains an integral part of fine-grained recognition, where it acts as an intermediary step with or without part-level supervision [9, 25, 36, 48, 49, 51, 67, 77, 79, 81–83, 85, 89, 90]. However, all these methods require joint training with image labels and focus mostly on discovering the most informative (discriminative) regions to ultimately help with the classification task.

Unlike previous methods as well as existing supervised methods that learn from annotated part segments [38, 50, 70, 75], our goal is to discover independent and semantically consistent parts without image-level or part-level labels . Bau et al. [2] and Gonzalez-Garcia et al. [28] inspect the hidden units of convolutional neural networks (CNNs) trained with image-level supervision (e.g., on ImageNet [61]) to understand whether part detectors emerge in them systematically. This is done by measuring the alignment between each unit and a set of dense part labels, and as such, the availability of manual annotations is required for interpretation. Most related to our work, however, are approaches for unsupervised part segmentation [4, 14, 37, 47, 64]. Based on the observation of [2, 28] that semantic parts do indeed emerge in deep features, Collins et al. [14] propose to use non-negative matrix factorization to decompose a pre-trained CNN's activations into parts. Similar observations had been previously discussed in [64] for constructing part constellations models and in [76] for part detection via feature clustering. To learn part segmentations in an unsupervised manner, Braun et al. [4] propose a probabilistic generative model to disentangle shape and appearance, but focus mostly on human body parts. Lastly, closest to our work is SCOPS by Hung et al. [37]; SCOPS is a self-supervised approach for object part segmentation from an image collection of the same coarse category, e.g., birds. The authors propose a set of loss functions to train a model to output part segments that adhere to semantic, geometric and equivariance constraints.

Other recent methods [69, 87] use generative adversarial networks for few-shot part segmentation, while [24] discover parts without supervision by interacting with articulated objects, and [53, 62, 63, 80] from motion in videos.

**Self-supervised and contrastive learning.** In self-supervised learning, one typically aims to design pretext tasks [18, 27, 58, 59, 84] for pre-training neural networks; these tasks are constructed such that the model has to capture useful information about the data that leads to learning useful features. Contrastive learning has recently emerged as a promising paradigm in self-supervised learning in computer vision, with several methods [7, 11, 32–35, 42, 57, 71, 78] learning strong image representations that transfer to downstream tasks. The key idea in contrastive learning is to encode two similar data points with similar embeddings, while pushing the embeddings of dissimilar data further apart [30]. In absence of labels, most contrastive methods use heavy data augmentations to create different views of the same image to use as a positive pair and are trained to minimize different variants of the InfoNCE loss [71].

We instead follow an approach to contrastive learning that is more tailored to semantic part segmentation, *i.e.* taking into consideration the dense nature of this problem. Our method is thus also related to self-supervised learning of dense representations [39, 60, 72, 88]. As these methods learn

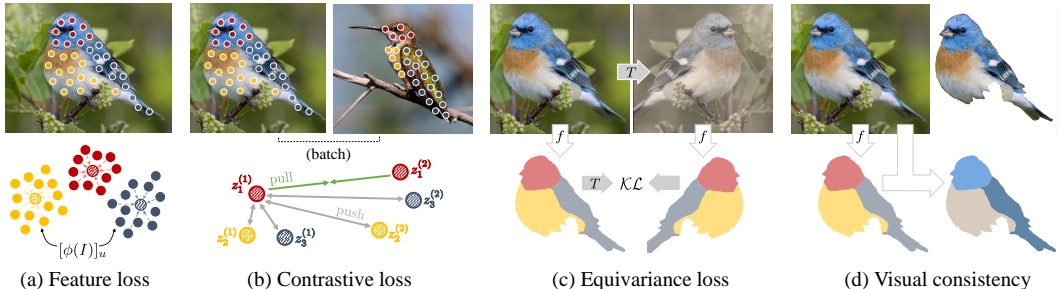

| (a) Feature loss | (b) Contrastive loss | (c) Equivariance loss | (d) Visual consistency |

Figure 1: **Training objectives.** We train our model with a set of loss functions that enforce several forms of consistency between the discovered parts. The feature loss a) ensures that parts are consistent within themselves. The contrastive loss b) discovers the same part in different images. The equivariance loss c) makes use of the fact that image transformations should not change part segmentations, and the visual consistency d) reconstructs a simplified version of the image from the parts to encourage visual consistency.

an embedding for every pixel, they cannot be directly applied for part segmentation and need either fine-tuning or a clustering step to produce part masks.

## 3 Method

Given a collection of images centered around a given type of objects (e.g., birds), we wish to automatically learn a part detector, assigning each pixel of the objects to one of $K$ semantic parts. Formally, we model the part segmentation task as predicting a mask $M \in \{0,1\}^{K \times H \times W}$ for an image $I \in \mathbb{R}^{3 \times H \times W}$, where $\sum_{k=1}^{K} M_u = 1$ for all pixels $u \in \{0, \dots, H-1\} \times \{0, \dots, W-1\}$. The mask thus assigns each pixel $u$ to one of $K$ parts and the part segmenter is a function $f : I \mapsto M$, implemented as a deep neural network, that maps an image $I$ to its part mask $M$. The mask is relaxed and computed in a differentiable manner, by applying the softmax operator at each pixel.

Since we are tackling this task without supervision, we have to construct a proxy task that will enforce $f$ to learn a meaningful decomposition of the image into its parts without the need for labelled examples. The rest of the section defines this task.

### 3.1 Contrastive feature discovery

Following prior work [14, 37], our primary cue for discovering parts is a deep feature extractor $\phi$, obtained as a neural network pre-trained on an off-the-shelf benchmark such as ImageNet, with or without supervision. In order to obtain repeatable and distinctive parts from these features, we propose to use a *contrastive formulation* [35, 71].

To this end, let $[\phi(I)]_u \in \mathbb{R}^d$ be the feature vector associated by the network to pixel location $u$ in the image. The idea is that, if pixel $v$ belongs to the same part type as $u$, then their feature vectors should be very similar when *contrasted* to the case in which $v$ belongs to a different part type. Since parts should be consistent irrespective of the particular object instance, comparisons extend within each image $I$, but also *across different images*. Thus, a naïve application of this idea would require a number of comparison proportional to the square of the number of pixels in the dataset, which is impractical.

Instead, we approach this issue by noting that contrastive learning would encourage features that belong to the same *part type* to be similar. This is even more true for features that belong to the same part *occurrence* in a specific image. We can thus summarize the code for part $k$ in image $I$ via an average part descriptor $z_k \in \mathbb{R}^d$:

$$z_k(I) = \frac{1}{|M_k|} \sum_{u \in \Omega} M_{ku} [\phi(I)]_u, \quad |M_k| = \sum_{u \in \Omega} M_{ku}, \tag{1}$$

where $\Omega$ represents all foreground pixels in the image. We can then *directly* enforce that pixels within the same part occurrence respond with similar features by minimizing the variance of descriptors

within the part:

$$\mathcal{L}_f(M) = \sum_{k=1}^{K} \sum_{u \in \Omega} M_{ku} \|z_k(I) - [\phi(I)]_u\|_2^2. \tag{2}$$

By doing so, we gain two advantages. First, pixels are assigned to the same part occurrence if they have similar feature vectors, as contrastive learning would do. Second, the part occurrence is now summarized by a single average descriptor vector $z_k(I)$ which has a differentiable dependency on the mask. Next, we show how we can express the rest of the contrastive learning loss as a function of these differentiable part occurrence summaries.

To this end, we use a random set (e.g., the mini-batch) of other images. Intuitively, we would like to maximize the semantic similarity between all the $k$-th parts *across* images and analogously minimize the semantic similarity between all other parts. This score is computed over a batch of $N$ images, each with $K$ descriptors $z_k^{(n)}$, where $n$ indexes the image. To reduce the number of comparisons, for each part $k$ we randomly choose a *target* $\hat{z}_k^{(n)} \in \{z_k^{(i)}\}_{i \neq n}$ out of the $N - 1$ other part $k$ occurrences in the batch.[1] With this, the contrastive loss can be written as usual:

$$\mathcal{L}_c = -\sum_{n=1}^{N} \sum_{k=1}^{K} \log \frac{\exp(z_k^{(n)} \cdot \hat{z}_k^{(n)} / \tau)}{\exp(z_k^{(n)} \cdot \hat{z}_k^{(n)} / \tau) + \sum_{j \neq k} \sum_{i \neq n} \exp(z_k^{(n)} \cdot z_j^{(i)} / \tau)}, \tag{3}$$

where $\tau$ is a temperature hyper-parameter that controls the "peakyness" of the similarity metric.

Note that, while this score function resembles the typical contrastive formulation in current self-supervised approaches, instead of generating the target $\hat{z}_k^{(n)}$ as an augmentation of the original image, here we can actually use a different image, since part $k$ has the same semantic meaning in both images. This formulation implicitly encourages two properties. On one hand, it maximizes the similarity of the *same* part type across images, and on the other hand, it maximizes the dissimilarity of *different* part types in the same and other images.

## 3.2 Visual consistency

While semantic consistency of part features is an important learning signal, these feature maps are of low spatial resolution and do not accurately align to image boundaries. We suggest that an effective remedy is to look for the *visual* consistency of the part itself. We can in fact expect most part occurrences to be characterized by a homogeneous texture. Generative modelling can then be used to assign individual pixels to different part regions based on how well they fit each part appearance.

This signal is in part complementary to feature-based grouping. As shown in Figure 2, clustering features from successive layers of a VGG-19 network [65] (pre-trained on ImageNet), when the receptive field of the features straddles two or more parts, grouping may sometimes highlight self-similar structures such as region boundaries instead of parts. On the other hand, image pixels can almost always uniquely be attributed to a single part.

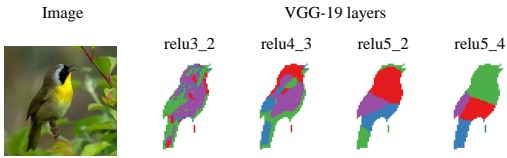

Figure 2: $K$-means clustering ($K = 4$) on foreground pixels for features extracted at different layers of a VGG-19 [65].

In our experiments, we show that even the simplest possible generative model, which assumes that pixels are i.i.d. samples from identical Gaussians, helps improving the consistency of the discovered parts. The negative log likelihood of parts under this simple model is given by the loss:

$$\mathcal{L}_v(M) = \sum_{k=1}^{K} \sum_{u \in \Omega} M_{ku} \left\| I_u - \frac{1}{|M_k|} \sum_{v \in \Omega} M_{kv} I_v \right\|_2^2. \tag{4}$$

This encodes the inductive bias that parts are roughly uniformly colored. Encouraging the model to learn parts that align with image boundaries. While more complex generative models can be used here, in our experiments (Table 2), this simple assumption already improved the results considerably.

---

[1]Note that samples are taken with respect to part occurrences, which are fixed, not with respect to the assignment of pixels to the parts (which are learned as the mask $M$). As a consequence, we do *not* need to differentiate through this sampler.

### 3.3 Transformation equivariance

Finally, we make use of the fact that an image transformation should not change the assignment of pixels to parts. We thus sample a random image transformation $T$ and minimize the symmetrized Kullback-Leibler divergence $\mathcal{KL}$ between the original mask and the mask predicted from the transformed image

$$\mathcal{L}_e(I, T(I)) = \sum_{u \in \Omega} \mathcal{KL}\left(T_u(f(I)), f_u(T(I))\right) + \mathcal{KL}\left(f_u(T(I)), T_u(f(I))\right) . \tag{5}$$

Here the $\mathcal{KL}$ divergence is computed per pixel, using the fact that the model predicts, via the softmax operator, a probability distribution over possible parts at each image location. This objective encourages commutativity of the function $f$ with respect to the transformation as it is minimized if $T(f(I)) = f(T(I))$, in other words, on equivariance under image transformations.

Note that, for equivariance, we need to define the action of $T$ on both the input image $I$ and the output $f(I)$. We consider simple random geometric warps (affine), which are applicable to any image-like tensor (thus even the pixels-wise predictions $f(I)$). We also consider photometric augmentations (e.g., color jitter), whose corresponding action in output space is the identity, because we wish the network to learn to be invariant to these effects (they do not change the part identity or location).

**Overall objective.** We learn $f$ by minimizing the weighted sum of the prior losses: $\lambda_f \mathcal{L}_f + \lambda_c \mathcal{L}_c + \lambda_v \mathcal{L}_v + \lambda_e \mathcal{L}_e$.

## 4   Experiments

In the following we validate our approach on three benchmark datasets, the Caltech-UCSD Birds-200 dataset (CUB-200-2011) [74], the large-scale fashion database (DeepFashion) [54] and PASCAL-Part [12]. Details regarding the datasets are given in the appendix. We carry out ablation experiments to study (a) the importance of the proposed objective functions, and (b) the role of supervised vs. unsupervised pre-training for the different components of our model. Lastly, we show that our method compares favorably to prior work both quantitatively and qualitatively.

**Implementation details.**   We model $f$ as a deep neural network, specifically a DeepLab-v2 [10] with ResNet-50 [31] as backbone, as it is a standard architecture for semantic image segmentation. Following SCOPS [37] we choose VGG19 [65] as the perceptual network $\phi$ and use ground truth foreground masks during training. Unless otherwise specified the backbone and perceptual network are pre-trained on ImageNet with image-level supervision. The perceptual network is kept fixed, *i.e.* its parameters are not further updated during training for part segmentation.

We use the same set of hyper-parameters for both, CUB-200 and Deep-Fashion, whereas some small changes are necessary for PASCAL-Part since the images are in a different resolution which typically impacts the magnitude of feature-based losses. We provide all implementation details in the appendix.

### 4.1   Evaluation Metrics

Prior work on unsupervised part segmentation [37] compares against unsupervised landmark regression methods [68, 86], due to the similarity between the two tasks and the limited availability of annotations. To do so, landmarks are obtained from part segmentations by taking the center of each mask, followed by fitting a linear regression model to map the predicted to the ground truth landmarks. We begin by taking a critical look at this evaluation metric.

We evaluate several baselines on CUB-200-2011 — namely, using the image midpoint, the center of ground truth keypoints and a single selected ground truth keypoint — and find that the landmark regression error does not correlate well with *segmentation* performance. For example, if we assume a model can accurately predict one *single* keypoint and nothing else (in this case the "throat"), the keypoint regression error is already lower than the previous state of the art. This means that a model that predicts one good part and $K - 1$ random parts would already outperform all previous methods. Thus, the metric does not sufficiently measure the segmentation aspect of the task, which is the main goal of our method, as well as that of [14, 36, 37].

Instead, we propose to measure the information overlap between the predicted labelling and the ground truth with Normalized Mutual Information (NMI) and Adjusted Rand Index (ARI) as we find

Table 1: **Comparison to prior work on CUB-200-2011 [74].** We report keypoint regression error as the normalized L2 distance (%), as well as (FG-)NMI and (FG-)ARI metrics. All methods predict $K = 4$ parts. $^\dagger$ uses image-level supervision.

| Method | Keypoint Regression Error ↓ | | | | FG-NMI↑ | FG-ARI↑ | NMI↑ | ARI↑ |
|---|---|---|---|---|---|---|---|---|
| | CUB-001 | CUB-002 | CUB-003 | CUB-all | | | | |
| Image midpoint | 27.3 | 26.7 | 27.2 | 23.5 | 0.0 | 0.0 | 0.0 | 0.0 |
| GT keypoint avg | 20.9 | 22.4 | 19.9 | 17.9 | 0.0 | 0.0 | 0.0 | 0.0 |
| "throat" kpt only | 16.4 | 14.9 | 15.2 | 12.1 | 11.6 | -16.2 | 4.6 | -8.3 |
| ULD [68, 86] | 30.1 | 29.4 | 28.2 | - | - | - | - | - |
| DFF [14] | 22.4 | 21.6 | 22.0 | - | 32.4 | 14.3 | 25.9 | 12.4 |
| SCOPS [37] (paper) | 18.5 | 18.8 | 21.1 | - | - | - | - | - |
| SCOPS [37] (model) | 18.3 | 17.7 | 17.0 | 12.6 | 39.1 | 17.9 | 24.4 | 7.1 |
| Huang and Li [36]$^\dagger$ | 15.1 | 17.1 | 15.7 | 11.6 | - | - | 26.1 | 13.2 |
| Ours | **11.3** | **15.0** | **10.6** | **9.2** | **46.0** | **21.0** | **43.5** | **19.6** |

this does not suffer from this drawback. Comparing to Intersection-over-Union (IoU) — which is commonly used to evaluate segmentation and detection performance — in an unsupervised setting, NMI and ARI have the advantage that they do not require the ground truth annotation to align exactly with the discovered parts and do not impose a constraint in the value of $K$, *i.e.* it does not need to be the same as the number of annotated categories. We propose to compute NMI and ARI not only on the full image, but also on foreground pixels only (FG-NMI, FG-ARI). The latter are stricter metrics that place the focus on *part* quality, dampening the influence of the background, which can be usually predicted with high accuracy using state-of-the-art segmentation or saliency methods. Importantly, these metrics can be computed even if a subset of the pixels are annotated in the dataset, and in particular even if only keypoint annotations are available, as in the case of CUB-200-2011.

### 4.2 Ablation Experiments

In Table 2 we evaluate the different objectives used to train our model. We first deactivate each loss and measure the impact it has on performance in two datasets, CUB-200-2011 and DeepFashion. Interestingly, we find that the different components differ in importance across the two datasets, even though we use the same hyper-parameters for both. On CUB-200-2011, the most important component is to enforce consistency within parts, whereas on DeepFashion visual consistency appears to have the largest impact. This likely comes from the different nature of "parts" in these two datasets. For birds, the parts are conceptually defined by shape, function and deformation (which is captured by features), whereas for the fashion dataset, parts such as T-shirts and trousers can be identified by their consistent color and texture (which is better captured by the image). Nonetheless, to achieve maximum performance both components are necessary in both datasets, as well as the equivariance and contrastive terms. To better understand the importance of the contrastive formulation, we replace it with a simple $\mathcal{L}_2$ loss, *i.e.* comparing part feature vectors $z_k$ *across* samples in the batch. We refer to this variant as "$\mathcal{L}_2$ instead of contrastive" and note that it performs significantly worse than the full model with the contrastive loss. To analyze the effect of using different images, we also train a model where we use parts in differently augmented versions (as is common in representation learning) of the *same* image instead ("$\mathcal{L}_c$ w/ different views"). Exploiting the information in different images leads to better performance. Finally, we establish a simple baseline by clustering perceptual features of concatenated layers `relu5_2`, `relu5_4` from a VGG19 (same layers as used in [37]) with $K$-means ($K = 4$). This simple clustering baseline performs quite well and almost reaches the performance of previous methods (Table 1), but the proposed approach is clearly stronger. Notably, feature clustering results in weaker performance for DeepFashion, which intuitively also explains why within-part consistency ($\mathcal{L}_f$) is not the most critical component for this dataset.

### 4.3 Eliminating Supervision

The method we have presented is unsupervised with respect to part annotations. However, similar to previous work [36, 37], we still rely on backbones pre-trained with ImageNet supervision (IN-1lk), and foreground-background segmentation masks. In Table 3 we remove these remaining, weakly

Table 2: **Ablation.** We remove various parts of our model and measure the decrease in performance. Additionally, we evaluate a baseline where we cluster VGG19 features unsing $k$-means.

| | | CUB-200-2011 (kp) | | DeepFashion (fg) | |
|---|---|---|---|---|---|
| **Variant** | | **FG-NMI** | **FG-ARI** | **FG-NMI** | **FG-ARI** |
| $k$-means cluster (VGG19) | `[relu5_2, relu5_4]` | 34.9 | 14.7 | 30.3 | 21.4 |
| w/o consistency within parts | $(\lambda_f = 0)$ | 29.7 | 11.7 | 40.3 | 40.0 |
| w/o consistency across parts | $(\lambda_c = 0)$ | 41.3 | 19.0 | 39.0 | 40.1 |
| w/o visual consistency | $(\lambda_v = 0)$ | 38.5 | 17.9 | 31.3 | 25.2 |
| w/o equivariance | $(\lambda_e = 0)$ | 29.3 | 11.2 | 41.5 | 42.7 |
| $\mathcal{L}_2$ instead of contrastive | $\mathcal{L}_c = \|z_k^{(n)} - \hat{z}_k^{(n)}\|_2^2$ | 34.0 | 13.4 | 36.7 | 32.0 |
| $\mathcal{L}_c$ w/ different views | | 44.4 | 20.2 | 36.4 | 33.4 |
| Ours | (full model) | **46.0** | **21.0** | **44.8** | **46.6** |

Table 3: **Elimination of supervision.** While our model is unsupervised with respect to part annotations of any form, we analyze its performance when moving from weight initialization with supervised models to weights from unsupervised models. The ablation is shown for $K = 4$ parts on CUB-200-2011 [74].

| Backbone of $f$ | Perceptual Network $\phi$ | FG Mask | FG-NMI | FG-ARI |
|---|---|---|---|---|
| ResNet50 (IN-1k supervised) | VGG19 (IN-1k supervised) | GT | 46.0 | 21.0 |
| ResNet50 (IN-1k supervised) | VGG16 (IN-1k supervised) | GT | 39.7 | 19.1 |
| ResNet50 (SwAV[7]) | VGG16 (IN-1k supervised) | GT | 35.4 | 16.4 |
| ResNet50 (SwAV[7]) | VGG16 (DeepCluster-v1 [6]) | GT | 32.3 | 14.0 |
| ResNet50 (SwAV[7]) | VGG16 (DeepCluster-v1 [6]) | [56] | 31.9 | 14.9 |
| ResNet50 (IN-1k supervised) | ViT (DiNO[8]) | GT | 43.9 | 19.7 |
| ResNet50 (SwAV[7]) | ViT (DINO [8]) | [56] | 42.7 | 20.0 |

supervised components step by step and replace them with unsupervised models. We notice, that none of the recent self-supervised methods provides models based on VGG architectures [65], although VGG is considered a much better architecture for perceptual-type losses than ResNet [31]. We thus use a VGG16 from DeepCluster-v1 [6]. For a fair comparison we directly compare to a supervised VGG16 and not our final model that uses VGG19. We find that the performance is indeed impacted by changing from supervised to unsupervised visual features ($-6$ NMI) and by replacing the supervised backbone ($-5$ NMI). But the final performance is still competitive with previous methods such as DFF [14] that use ImageNet supervision and masks. Using an unsupervised saliency method [56] for segmentation instead of ground truth foreground masks only causes a negligible drop in performance. We further experiment with a self-supervised vision transformer(ViT) [8] for $\phi$. Following [1], we extract the dense key features of the last transformer block of DINO [8] network with stride 4. We observe that self-supervised ViT features perform significantly better than self-supervised CNN features. When we remove all supervisions, the performance drop is minimal ($-1$ NMI).

## 4.4 Comparisons with the State of the Art

**CUB-200.** On CUB-200 (Table 1 and Figure 3), we evaluate keypoint regression performance to be directly comparable to previous work. Due to the aforementioned limitations of this metric, we also evaluate NMI and ARI on both the foreground object (denoted with **FG**) and the whole image. We use the publicly available checkpoint of SCOPS [37] to compute these new metrics for their method. Additionally, we run DFF [14] using their publicly available code. Finally, we are even able to improve over [36] who use class labels for fine-grained recognition during training.

**DeepFashion.** Finally, in Table 4 and in Figure 4 we evaluate our method on the DeepFashion dataset, reporting (FG-)NMI and (FG-)ARI scores for $K = 4$ predicted parts. Our model is able to identify more meaningful parts (namely: hair, skin, upper-grament, lower-garment) than SCOPS.

Table 4: **DeepFashion dataset.** We compute (FG-)NMI and (FG-)ARI for $K = 4$ parts.

| | FG-NMI | FG-ARI | NMI | ARI |
|---|---|---|---|---|
| SCOPS [37] | 30.7 | 27.6 | 56.6 | 81.4 |
| Ours | **44.8** | **46.6** | **68.1** | **90.6** |

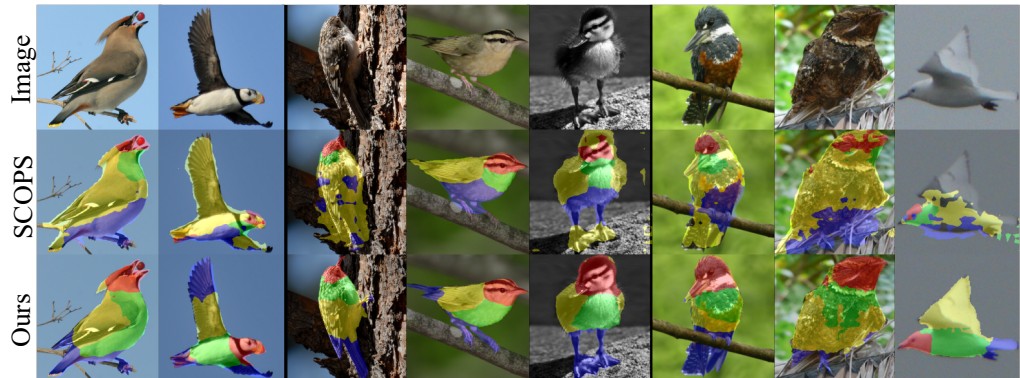

Figure 3: **CUB-200 dataset.** Qualitative examples for SCOPS [37] and our method show that our model is able to find clearer part boundaries even in difficult poses, e.g., open wings.

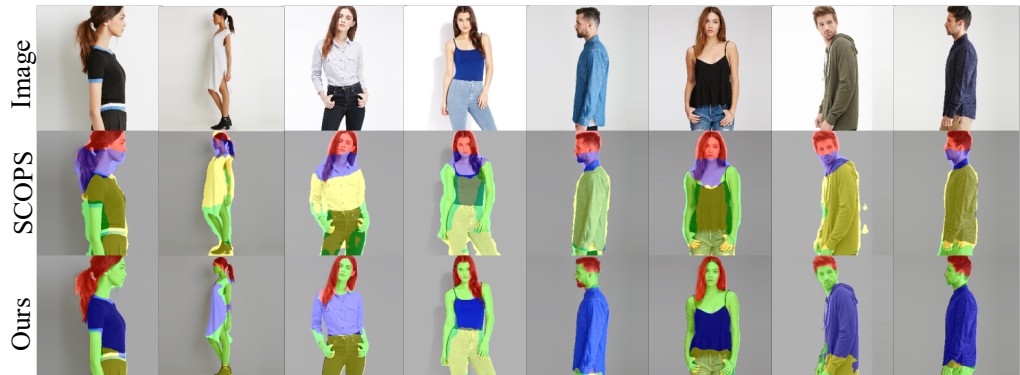

Figure 4: **DeepFashion dataset.** Our model is able to separate the hair from the rest of the head and correctly finds the boundary between upper and lower garments.

Table 5: **PASCAL-Part dataset.** We show NMI and ARI scores on individual classes in pascal parts [13]. All methods predict $K = 4$ parts.

| | NMI | | | | | | | | | | ARI | | | | | | | | | |
|---|---|---|---|---|---|---|---|---|---|---|---|---|---|---|---|---|---|---|---|---|
| | sheep | horse | cow | mbike | plane | bus | car | bike | dog | cat | sheep | horse | cow | mbike | plane | bus | car | bike | dog | cat |
| DFF [14] | 12.2 | 14.4 | 12.7 | 19.1 | 16.4 | 13.5 | 9.0 | 17.8 | 14.8 | 18.0 | 21.6 | 32.3 | 23.3 | 37.2 | 38.3 | 28.5 | 24.1 | 39.1 | 32.3 | 37.5 |
| SCOPS [37] | 26.5 | 29.4 | 28.8 | 35.4 | 35.1 | 35.7 | 33.6 | 28.9 | 30.1 | 33.7 | 46.3 | 55.7 | 51.2 | 59.2 | 68.0 | 66.0 | 67.1 | 52.4 | 52.2 | 46.6 |
| $K$-means | 34.5 | 33.3 | 33.0 | 38.9 | 42.8 | 37.5 | **38.4** | 35.2 | 40.4 | 44.2 | 58.3 | 66.8 | 59.0 | 63.1 | 76.8 | 66.4 | 70.6 | 63.2 | 70.2 | 71.9 |
| Ours | **35.0** | **37.4** | **35.3** | **40.5** | **45.1** | **38.8** | 36.8 | **34.8** | **46.6** | **47.9** | **59.8** | **68.9** | **59.7** | **64.7** | **79.6** | **67.6** | **72.7** | **64.7** | **73.6** | **75.4** |

**PASCAL-Part.** To understand the applicability of the method to a wide variety of objects and animals, we also evaluate on the PASCAL-Part dataset in Table 5 and Figure 5. We train one model per object class, as in [37]. However, we note that Hung et al. [37] only evaluate foreground-background segmentation performance in their paper. We thus train their method on each class and report NMI and ARI for quantitative comparisons. For DFF we perform non-negative matrix factorization on the set of features of each class separately. Finally, we consider an interestingly strong baseline, *i.e.* $K$-means clustering on foreground pixels trained for each class separately. We find that this baseline even outperforms prior work [14, 37] by a significant margin. However, our method strikes a balance between feature similarity and visual consistency, achieving superior part segmentation to $K$-means as well as previous methods. One possible explanation for the disadvantage of SCOPS and DFF to simple $K$-means clustering is that they learn foreground-background separation and discover semantic parts in the foreground simultaneously, which appears to be sub-optimal for either task, *i.e.* it harms both the foreground and the part segmentation, as also seen in Figure 5.

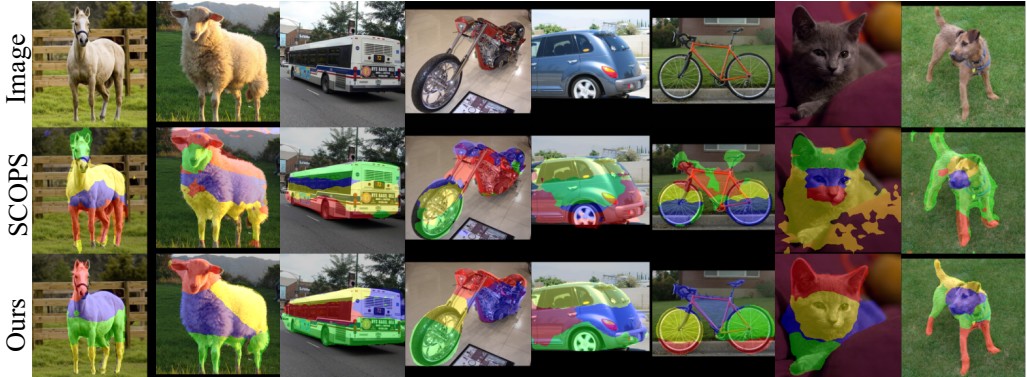

Figure 5: **PASCAL-Part dataset.** We train one model per class for both, our model and SCOPS [37]. For animals we find are able to separate different body parts. (More examples in the appendix.)

## 5    Discussion

**Limitations.**    One possible drawback of our approach is that there is no underlying reason why parts of an object should have uniform appearance, as enforced by our visual consistency objective (e.g., the wheels of a car or a striped garment). However, our main assumption is that different occurrences of the same part (e.g., the mouth for two people) are more similar to each other than two different parts (e.g., an eye and a mouth). While this assumption is of course not universally applicable it is true often enough to be helpful and, complemented by the other objectives, it leads to a considerable performance improvement over prior work that does not use this concept. Although we experimented with more complex visual modelling (e.g., higher level statistics or textures), this did not yield meaningful improvements and is thus left for future investigation. Another limitation is that parts discovered in a self-supervised manner might not necessarily agree with expected labels or human intuition. A critical control parameter for this is the number of parts $K$. It controls the granularity of the part segmentation and is left as a hyper-parameter since it is up to the user to decide the level of decomposition. For example, for humans one could segment arms, legs, torso and head (five parts) or decompose arms into hands, fingers, etc. In the appendix we show results of our method for different $K$. Finally, the main failure mode of the current model is failing to separate the foreground from the background which leads to messy segmentations and scrambled masks (see the appendix qualitative examples).

**Broader Impact.**    Supervised learning often requires highly-curated datasets with expensive, time-consuming, manual annotations. This is especially true for pixel-level tasks (e.g., segmentation) or tasks that require expert knowledge (e.g., fine-grained recognition). As a result, increasing attention is being placed on improving image understanding using little or no supervision. Since part segmentation datasets are limited in number and size, a direct positive impact of our approach is that discovering semantic object parts in a self-supervised manner can significantly increase the amount of data that can be leveraged to train such models. Finally, as with all methods that learn from data — and especially in the case of self-supervised learning — it is likely that underlying biases in the data affect the learning process and consequently predictions made by the model.

## 6    Conclusion

We have proposed a self-supervised method for discovering and segmenting object parts. We start from the observation, also discussed in prior work [2, 14], that deep CNN layers respond to semantic concepts or parts and thus clustering activations across an image collection amounts to discovering dense correspondences among them. We further expand upon this idea by introducing a contrastive formulation, as well as equivariance and visual consistency constraints. Our method relies only on the availability of foreground/background masks to separate an object of interest from its background. However, as we show experimentally, it is possible to leverage unsupervised saliency models to acquire such masks, which allows for a model that has no supervised components at all.

## Acknowledgements and Funding Disclosure

S. C. is supported by a scholarship sponsored by Facebook. I. L. is supported by the European Research Council (ERC) grant IDIU-638009 and EPSRC VisualAI EP/T028572/1. C. R. is supported by Innovate UK (project 71653) on behalf of UK Research and Innovation (UKRI) and ERC grant IDIU-638009. A. V. is supported by ERC grant IDIU-638009. We thank Luke Melas-Kyriazi for providing precomputed masks for [56].

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
