# Appendix: Unsupervised Part Discovery from Contrastive Reconstruction

**Subhabrata Choudhury**     **Iro Laina**     **Christian Rupprecht**     **Andrea Vedaldi**

Visual Geometry Group
University of Oxford
Oxford, UK
subha,iro,chrisr,vedaldi@robots.ox.ac.uk

## 1   Datasets

**CUB-200-2011.**   The Caltech-UCSD Birds-200 [14] is a dataset for fine-grained recognition, comprising of 11,788 images of 200 bird species. The dataset provides keypoint annotations for 15 part locations and a foreground segmentation mask per image. We use the official train split for training our model. (Dataset license: none; copyrights for individual images remain with their creators. User agreement: none)

**DeepFashion.**   DeepFashion (In-Shop Clothes Retrieval Benchmark) [10] is a fashion dataset containing 52,712 densely labelled images of people in different clothing items. The labels include 15 categories and a background class. We use the entire dataset and train on the official train split. For evaluation, we combine query and gallery splits (6922 images). (Dataset license: non-commercial research purposes only; images are not property of the dataset creators. User agreement: yes)

**PASCAL-Part.**   PASCAL-Part [2] is an extension of the PASCAL VOC 2010 dataset [5] providing part level annotations for the 20 categories. In total, the dataset contains 10,103 training and validation images and 9,637 testing images. We train a single model for each of the following 10 categories: sheep, horse, cow, motorbike, plane, bus, car, bike, dog, cat. We only consider images for which the respective object category occupies at least 20% of the image (measured by its bounding box) during training and evaluation. (Dataset license: none; the dataset includes images obtained from Flickr — use of these images must respect the corresponding terms of use. User agreement: none)

## 2   Experiment Details

**Implementation details.**   We train our model with SGD using a learning rate of $10^{-5}$, weight decay of $5 \cdot 10^{-4}$, batch size of 6 and image size of $256 \times 256$. For the contrastive and feature objectives ($\mathcal{L}_f$ and $\mathcal{L}_c$) we use features from a VGG19 [13] pre-trained on ImageNet [12]. We empirically found that using layers `relu3_2` and `relu5_4` with weight factors 0.33 and 1 respectively works best here. brightness (±30For the equivariance loss ($\mathcal{L}_e$), we use the following image transformations: color jitter as in [8] — brightness ($\pm 30\%$), contrast ($\pm 30\%$), saturation ($\pm 30\%$), and hue ($\pm 30\%$) — random rotations ($\pm 60°$) and translations ($\pm 10\%$). For CUB-200-2011 and DeepFashion, we train our models using $\lambda_f = 5$, $\lambda_c = 2.3 \cdot 10^3$, $\lambda_v = 30$, $\lambda_e = 5.7 \cdot 10^3$. For PASCAL-Part we use $\lambda_f = 5$, $\lambda_c = 2.3 \cdot 10^3$, $\lambda_v = 30$, $\lambda_e = 5.7 \cdot 10^4$, *i.e.* higher equivariance. We performed Bayesian hyper-parameter sweeps to tune the weights using Weights & Biases.[1] For all datasets, we train our model on foreground pixels only, using ground truth foreground-background masks. To ensure fair comparisons, at test time we use binary masks predicted from a DeepLab-v2 with ResNet-50 [6] as backbone (same architecture as our part segmentation network), which is trained for foreground-background segmentation on each dataset. In Table 3 of the main paper, we have also presented a *fully*

---

[1] https://wandb.ai

*unsupervised* approach, using an unsupervised method [11] to obtain binary masks on CUB-200-2011 for both training and evaluating our method.

**Computation time.** We have used NVIDIA Tesla P40 and RTX 6000 GPUs to train the model. On a Tesla P40 it takes 2.2 days to train the model.

**Baselines.** We compare our model to prior work and a $K$-means baseline, for which we provide details below.

- **DFF [3]:** Following the original paper and publicly available code,[2] we apply non-negative matrix factorization to the activations of the last convolutional layer of VGG19 (`relu5_4`). The computed factors decompose an image into parts, thus the method can be used for part co-segmentation and, under certain conditions, is equivalent to spectral clustering [4]. DFF does not require training but needs to be applied over the whole test set at once during inference in order to ensure that the parts correspond semantically across samples.

- $K$**-means baseline:** A critical component of our method, as well as SCOPS and DFF, is deep feature similarity within semantic regions of an image/object. As such, we deem feature clustering a relevant baseline and perform $K$-means clustering[3] (with $K = 4$) on VGG19 features for each dataset. As in [8], we use concatenated features from layers `relu5_2` and `relu5_4` (combined feature dimension is 1024), which we found works best for $K$-means. We upsample the feature maps to $64 \times 64$ and normalize each feature vector to unit norm prior to clustering. We perform clustering on the foreground features only, using foreground masks predicted with a DeepLab-v2 [1] trained on each dataset. We run $K$-means for 100 steps and report the best out of 5 initializations. We find that $K$-means is a strong baseline and in some cases it outperforms prior work.

**Evaluation details.** For the evaluation on CUB-200-2011, we follow prior work [8] and compute the landmark regression error, *i.e.* we fit a linear regression model (using the training set) to map predicted keypoints to ground truth keypoints. Since our method produces segmentation masks, we use the mask centers as keypoints. For compatibility with previous work, we evaluate the fitted model on the test split provided by [9] on the first three classes (CUB-001, CUB-002, CUB-003). In addition, we report the landmark regression error for all 200 classes. We train a single linear regressor for all classes combined.

Due to problems with the keypoint error that we have discussed in the main paper, we advocate the use of different metrics for evaluating part segmentation in future work. The proposed metrics (FG-)NMI and (FG-)ARI can be computed on both keypoint (CUB-200-2011) or pixel-wise (DeepFashion, PASCAL-Part) annotations. We emphasize the difference between **FG**-NMI/ARI and NMI/ARI. For all datasets, we compute FG-NMI and FG-ARI on images cropped around the object of interest (using its bounding box) and on foreground pixels only. As a result, these metrics reflect the quality of part segmentation without any influence from the background. In addition, we compute NMI and ARI on the *full* image (*i.e.* uncropped, with its smaller size resized to 256 pixels, including the background pixels) for CUB-200-2011 and DeepFashion. Since on PASCAL-Part there can be multiple instances of the same category in one image, we also crop the image to the target objects' bounding boxes for NMI/ARI calculation (but evaluate *with* background pixels in this case). The metrics are computed on the corresponding test sets. On CUB-200-2011 we compute the FG scores on the test split provided by [9] (same as the landmark regression error). Full-image NMI and ARI are computed on the official CUB test set for fair comparisons with [7].

## 3 Additional Results

### 3.1 Number of parts

Since part discovery depends on the chosen $K$, we evaluate our model for different values, namely $K = 4, 6, 8$. We report quantitative results for CUB-200-2011 and DeepFashion in Table 1. We note that NMI is not comparable across different numbers of classes/parts. ARI is adjusted for chance and should thus be comparable across $K$. We show additional qualitative results for $K = 4$ in Figures 1 and 2 and for $K = 6, 8$ in Figure 3.

---

[2]https://github.com/edocollins/DFF
[3]using the the implementation from https://github.com/facebookresearch/faiss

Table 1: **Evaluation of different number of parts.** We show results for different values of $K$ on CUB-200-2011 and DeepFashion.

| Variant | CUB-200-2011 (kp) | | | | DeepFashion (fg) | | | |
|---|---|---|---|---|---|---|---|---|
| | FG-NMI | FG-ARI | NMI | ARI | FG-NMI | FG-ARI | NMI | ARI |
| $K = 4$ | 46.0 | 21.0 | 43.5 | 19.6 | 44.8 | 46.6 | 68.1 | 90.6 |
| $K = 6$ | 47.2 | 23.0 | 44.4 | 20.7 | 43.5 | 42.2 | 66.2 | 91.0 |
| $K = 8$ | 58.2 | 34.0 | 51.5 | 28.3 | 39.2 | 30.7 | 62.4 | 90.6 |

## 3.2 Variability in results

To obtain a robust measure for the performance of our method, we have trained our model with $K = 4$ multiple times (with 5 different random seeds) and report the mean and standard deviation in Table 2.

Table 2: **Variability in results.** We run our model with $K = 4$ on CUB-200-2011 and DeepFashion with 5 different seeds and report mean $\pm$ standard deviation of NMI and ARI.

| Dataset | FG-NMI | FG-ARI | NMI | ARI |
|---|---|---|---|---|
| CUB | $45.3 \pm 2.8$ | $20.5 \pm 1.5$ | $42.8 \pm 1.7$ | $19.2 \pm 0.5$ |
| DeepFashion | $44.6 \pm 0.4$ | $46.1 \pm 0.6$ | $68.2 \pm 0.2$ | $90.7 \pm 0.1$ |

## 3.3 Failure cases

In Figure 5 we show failure cases of our model on CUB-2011 dataset. In most of the examples we observe the leading reason of failure is the low quality of the predicted object mask.

## 3.4 Qualitative Results on Loss Ablation

We show additional qualitative results on CUB for loss ablationin Figure 6. We observe that when any of the losses is removed from the full model visual quality drops.

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

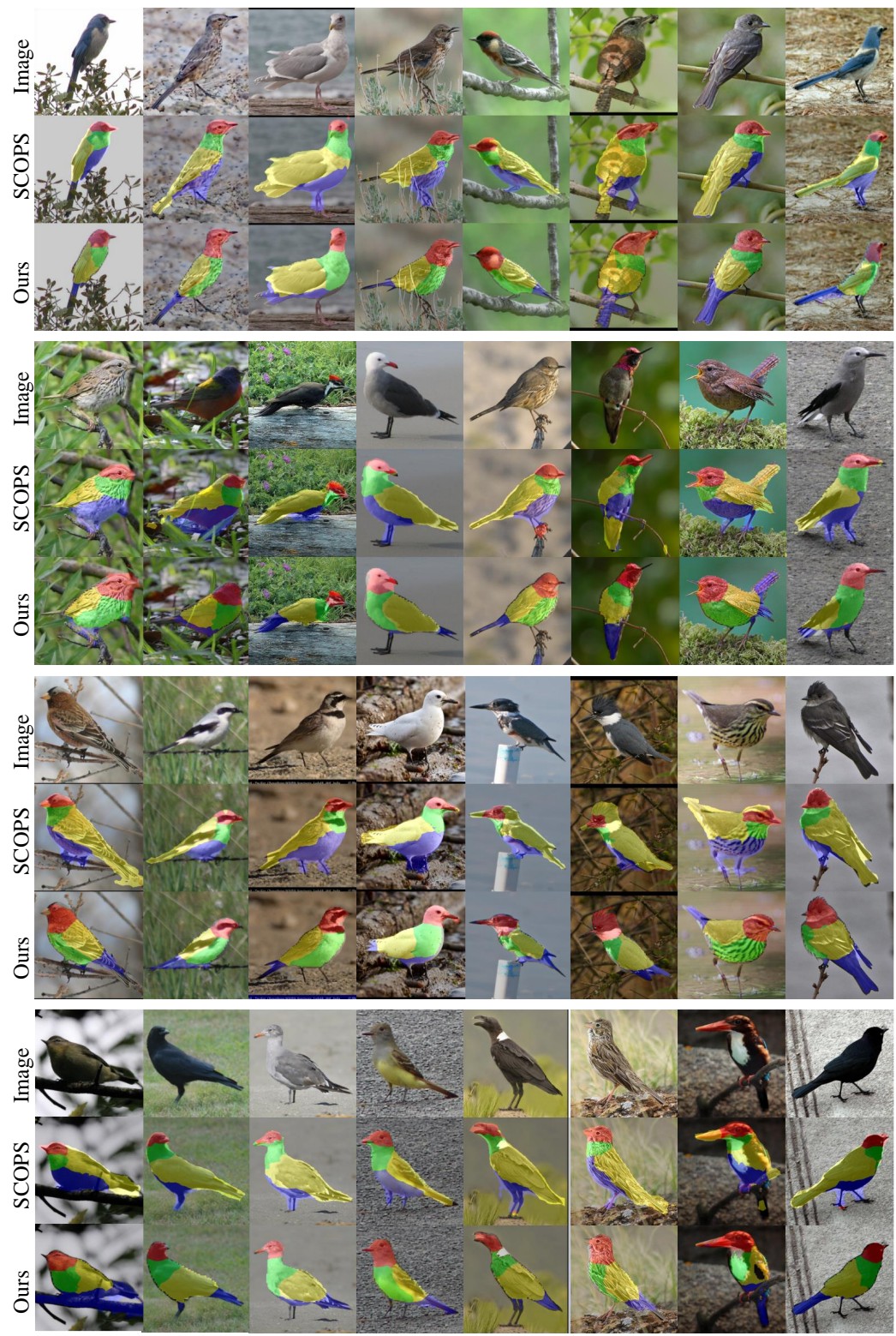

Figure 1: **CUB-200-2011 Dataset.** Additional qualitative examples for SCOPS [8] and our method.

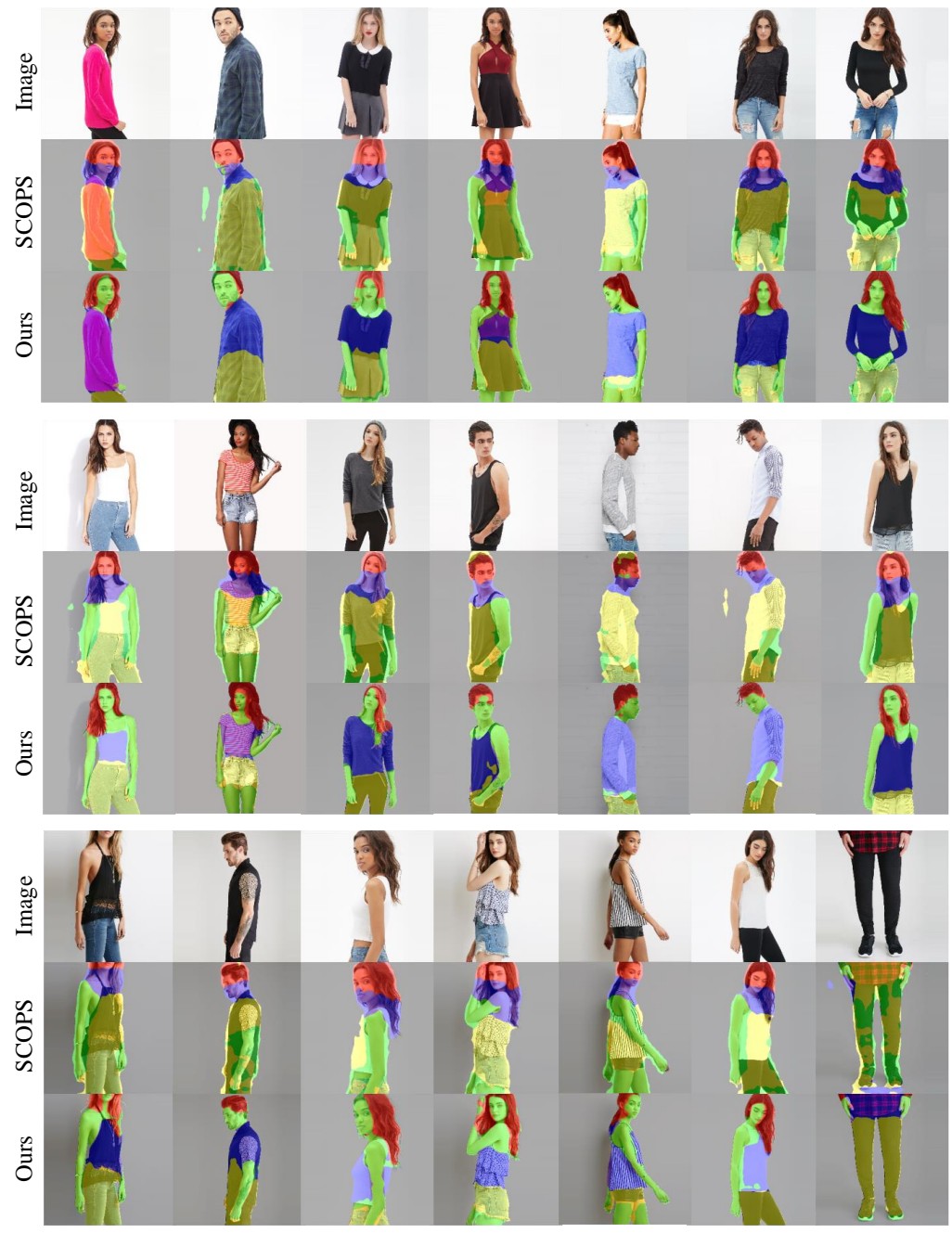

Figure 2: **DeepFashion Dataset.** Additional qualitative examples for SCOPS [8] and our method.

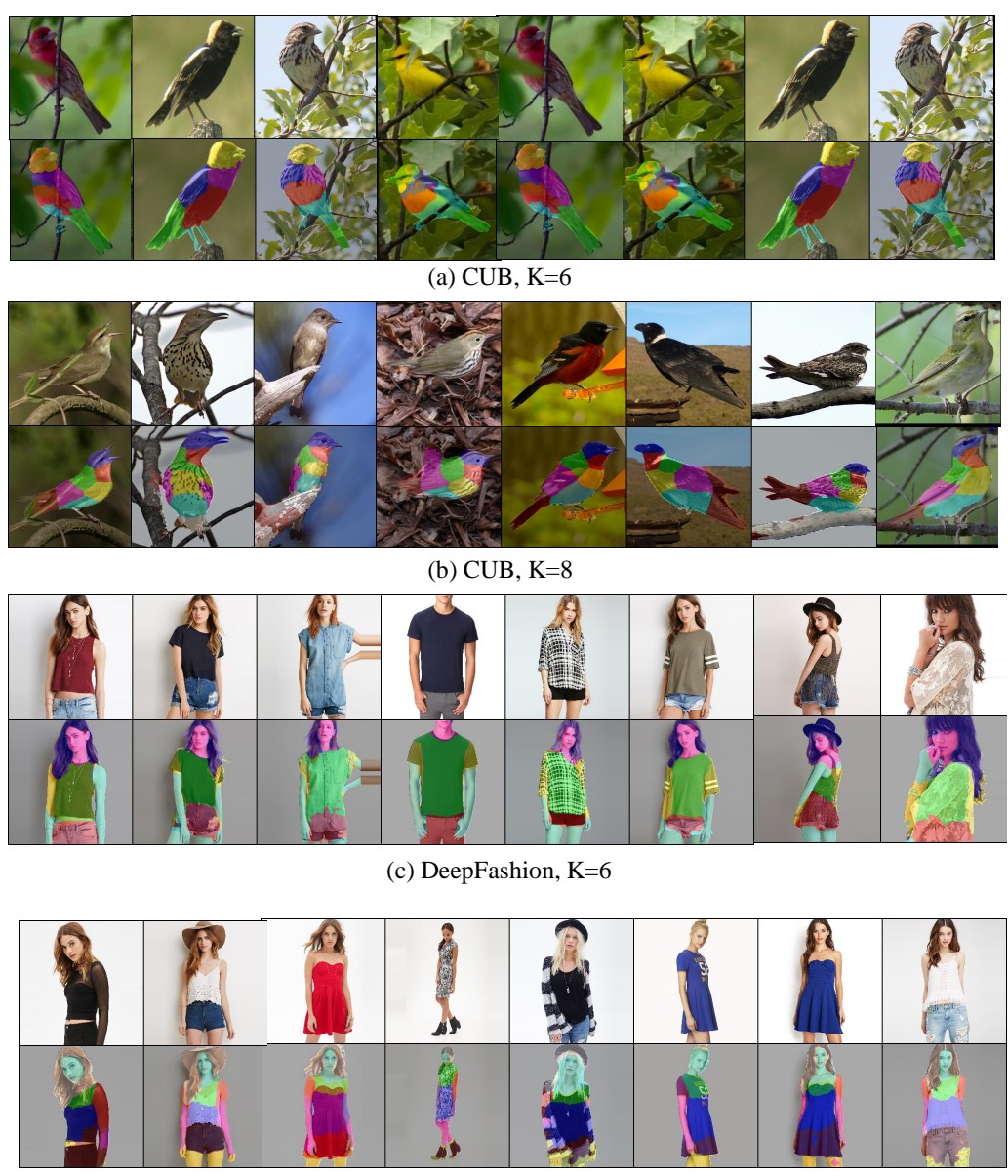

(a) CUB, K=6

(b) CUB, K=8

(c) DeepFashion, K=6

(d) DeepFashion, K=8

Figure 3: Qualitative examples of our model prediction for K=6 and K=8 on CUB-200-2011 and DeepFashion.

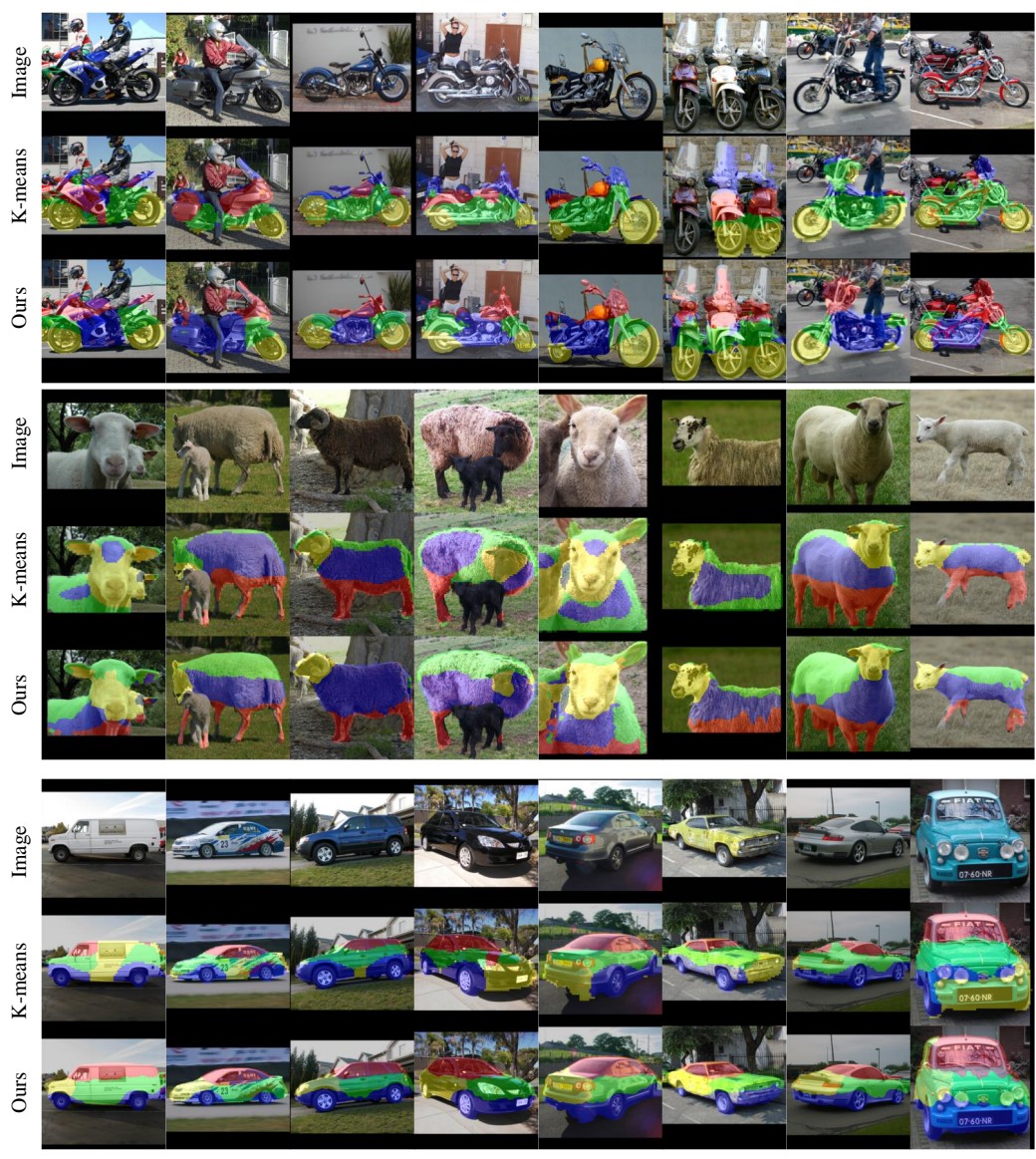

Figure 4: **PASCAL-Part Dataset.** Additional qualitative results for PASCAL-Part dataset for our model and the $K$-means baseline.

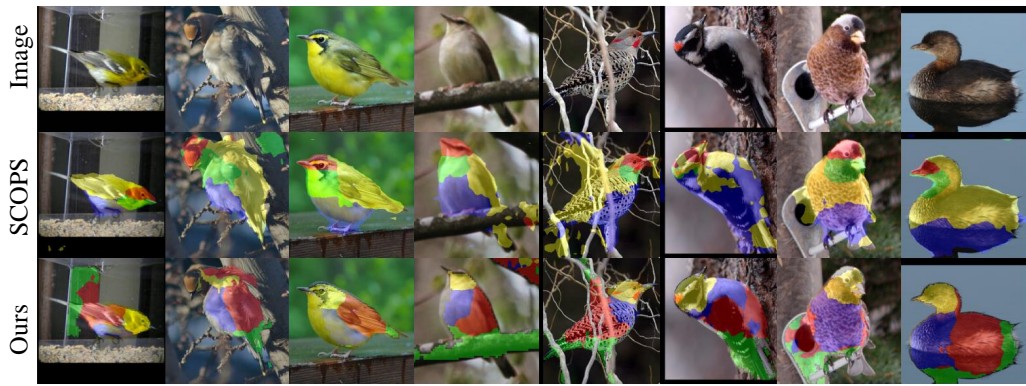

Figure 5: **CUB-200-2011 Dataset.** Qualitative examples of a failure mode for our model along with SCOPS [8] predictions. Most failures occur due to subpar background predictions.

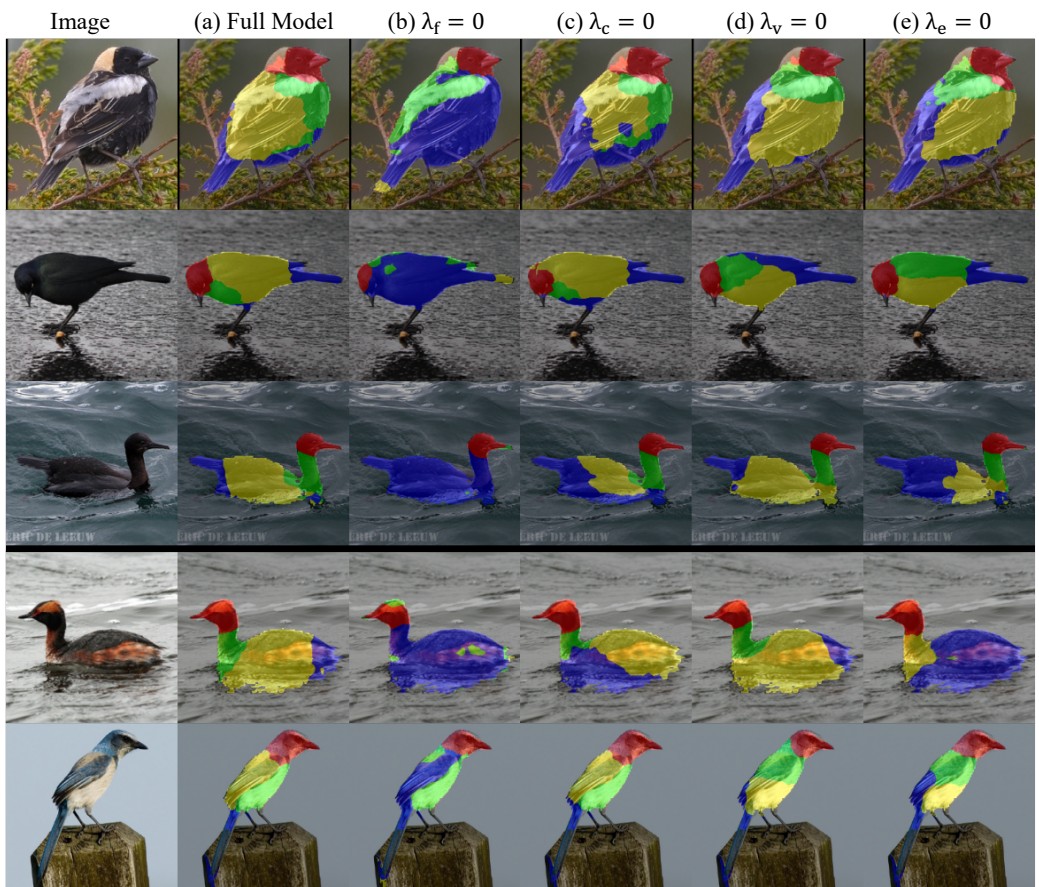

Figure 6: **CUB-200-2011 Dataset.** Qualitative examples of loss ablation. We show qualitative results for the (a) full model, when we remove one of (b) feature loss, (c) contrastive loss, (d) visual consistency loss, and (e) equivariance loss from our models predictions quality worsens.