# OpenReview forum: "Unsupervised Part Discovery from Contrastive Reconstruction"
_NeurIPS.cc/2021/Conference — NeurIPS 2021 Poster_

### Official Review · Reviewer_ogkV · 2021-07-16

**Rating:** 6
**Confidence:** 3

**Summary:**

The authors develop a method for segmenting an object into parts, given a dataset of images of that object category and foreground masks for each object. This method is a variant of contrastive self-supervised learning, which aims to make learned features within positive sample pairs more similar and features within negative sample pairs more disparate. In this case, the samples are feature vectors averaged over proposed part segmentation masks within each image, rather than feature vectors corresponding to an entire image as in now-standard approaches. Applying this principle, along with several additional contraints (auxiliary losses), leads to state-of-the-art performance on three semantic part segmentation benchmarks. The authors also introduce two new metrics for semantic part segmentation that they argue better capture the nature of the problem.

**Ethical Concerns:**

No, I don't have ethical concerns.

**Limitations And Societal Impact:**

Yes.

**Main Review:**

Figuring out how to represent objects with substructure (i.e., parts) is an important and difficult challenge; many of the things we interact with and depend on in everyday life can be decomposed into meaningful parts (e.g that perform different functions), and understanding this fact is central to how we perceive and use those objects. It also makes sense, to me, to tackle this problem with a self-supervised approach, because collecting part supervision labels for a wide range of objects across many categories seems hopeless in the long run: there are too many categories with distinct substructures, not all exemplars of a category have the same substructure, and in the case of man-made objects, new types of substructure will be created. Perhaps most importantly, the notion of a "part" is highly ambiguous.

My main reaction to this paper is surprise that the authors do not really consider the last of these difficulties -- neither in the introduction in talking about why their task is important and interesting, nor in the methods (which don't seem well-suited to handle this problem). They do get state-of-the-art results on what are apparently widely-used benchmarks, and they develop what seems to me a better set of metrics for assessing part-based segmentation given ground truth annotations. For these latter reasons I'd guess that this work might be valuable to people who work on the same problem, so I'm open to accepting the paper. But I hope that the authors can spend substantial space in the manuscript motivating their approach, addressing its limitations, and (ideally) modifying it to tackle the core problem I describe below -- or can explain why I'm wrong in thinking this is the core problem.

_The Core Problem: What makes a part a part?_ Based on the authors' introduction and explanation of their methods, the notion of "part" they seem to be going for here is some combination of "functional" (a part is something that performs a distinct function), "semantically conventional" (a part is something that most people agree is a part, regardless of the underlying reasons), and "appearance-based" (a part is something that has relatively uniform texture or color.) In particular, they seem to think that a learning method that implicitly uses the last of these notions (appearance-based) will suitably capture the first two (which is what they evaluate on.)

But it seems to me that in most cases, relatively uniform appearance isn't a good way to discover or think about parts of objects. There's no underlying reason why parts of an object that perform a distinct function should have uniform appearance that's different across different parts, and it's easy to think of examples where they don't: wheels of a car, wings of a butterfly, organs on a face (eyes, ears, nose, mouth), and so on. It's possible that, by virtue of supervised or self-supervised training, deep ANNs do learn higher-level feature spaces in which features are uniform within each part and distinct across parts -- but I haven't seen good evidence of this, and that doesn't mean that parts will have uniform texture at the pixel level. Indeed, the visual examples in the work here don't often match up with what I think of as a part (e.g. segmenting people into "hair" and "skin" rather than "head" and "arms").

Instead, the most natural notion of a part that **could** be learned from visual observations is one based on motion: a part is something that can move partly independently of the rest of an object. A functional definition might be closer to the truth in some cases (e.g. the porch of a house doesn't move independently of the rest of the house), but understanding the distinct functions of different spatial components of an object is probably beyond the scope of current computer vision methods; in any case, function and independent movability are highly correlated, as in the examples I listed in the paragraph above.

There are other recent works that _do_ recognize this notion of part as more fundamental than appearance, and develop methods that try to tackle it accordingly (whether not they succeed) -- for instance, [A,B]. In particular, the problem formulation in [A] is very similar to the one used here -- mainly the testing datasets are different.

Beyond the issue with the underlying notion of a part, I have another major concern with this approach: the number of parts will be different for different categories, and may also vary within a category, yet here the number of parts (K) is a fixed element of the architecture. This means that an architecture will have to be tailored for a specific category, depending on what the authors think is the "true" number of distinct parts. This goes against the principles of self-supervised learning, in which the designers of an algorithm should not need to know too much about the ground truth labels for a dataset to apply the approach. In other methods (like [A,B]), only the maximum number of parts is chosen in advance; different scenes may have different numbers of "occupied" slots.

It's also worth remarking that the decomposition of an object into parts is not fundamentally different from decomposition of a scene into objects -- unless there is some strong philosophical reason to treat "parts of an object" differently from "parts of a scene." Because of this, I'm surprised that the authors don't relate their work to unsupervised object discovery methods like [B,C,D] or similar.

Overall, I think this work would benefit from much stronger conceptual motivation and development, and ideally would attempt to tackle unsupervised part discovery in a more principled way -- I think this is possible with more or less the framework the authors use now if given some more thought. However, as far as I can tell, the quantitative improvements over competing methods are real and large, and it seems like it's more common in this research area to use the "conventional semantic" notion of an object part -- so addressing my conceptual concerns, even without running more experiments, would lead to a pretty strong submission overall.

[A]: Sabour, Sara, et al. "Unsupervised part representation by flow capsules." International Conference on Machine Learning. PMLR, 2021.

[B]: Bear, Daniel M., et al. "Learning physical graph representations from visual scenes." arXiv preprint arXiv:2006.12373 (2020).

[C]: Kipf, Thomas, Elise van der Pol, and Max Welling. "Contrastive learning of structured world models." arXiv preprint arXiv:1911.12247 (2019).

[D]: Hénaff, Olivier J., et al. "Efficient visual pretraining with contrastive detection." arXiv preprint arXiv:2103.10957 (2021).

**Time Spent Reviewing:**

3

---

> ### Author Response · Authors · 2021-08-10
> **Response**
>
> We thank the reviewer for the feedback and the deep analysis of the problem formulation. We agree that a theoretical grounding is important and we will include it in the introduction and discussion. We are grateful that the reviewer acknowledges that the proposed method is able to show good performance gains over existing methods in an established sub-field of unsupervised learning.
>
> Before addressing the “Core Problem”, we think it might be helpful to first analyze this statement:
>
> > *It's also worth remarking that the decomposition of an object into parts is not fundamentally different from decomposition of a scene into objects -- unless there is some strong philosophical reason to treat "parts of an object" differently from "parts of a scene."*
>
> While there is no universally accepted formal definition, the nature of objects and object parts is accepted as different. For example, for Gibson [Gibson66], an object is “detachable”, i.e. something that, at least conceptually, can be picked up and moved to a different place _irrespective of the rest of the scene_. Parts, in contrast, are constituent elements of an object, and cannot be removed without breaking the object, i.e. they are essential to the object and occur across most instances of the class. These fundamental differences lead to sufficiently different statistics and it is difficult to envision methods that can do both, i.e. methods that work well for scene decomposition into objects [MONet19, SlotAttention20, GENESIS20] do not do well for parts (e.g. [2, 10, 26, 43]) and vice-versa.
>
> Concretely:
>
> 1. Methods that discover objects usually do not also assign a type to them, whereas parts are always identified by type (e.g. head vs beak) in addition of instance;
> 2. The layout and presence of parts is usually constrained (e.g. the head is attached to the torso, and usually both or none appear)  whereas objects in a scene can be arranged more or less arbitrarily and in any number.
>
> Nevertheless, we have experimented with a representative method [SlotAttention20] on the CUB dataset. The discovered parts are orderless (by model design) resulting in NMI and ARI scores of 0 in quantitative evaluation as Slot Attention cannot identify correspondences across images. Qualitatively, the discovered part regions are not consistent across different birds (even when we account for orderlessness) and spatially disconnected regions exist in a single slot. As per the reviewer’s suggestion, we are happy to revise the paper to relate our work to the line of work on unsupervised scene decomposition and report and discuss the above findings.
>
>
> > *The Core Problem: What makes a part a part?*
>
>
> This is indeed a core problem, but so far no theoretical/philosophical consensus exists in the field. Moreover, the same problem exists for almost any type of concept one can think of learning. We cited [Gibson66] as an example definition of what “makes an object an object”, but that definition is still controversial and not entirely formal.
>
> In our case, we combine three simple learning principles as “part proxy”: consistency to transformation (equivariance) and  internally-consistent and distinctive appearance (see below). As for most of machine learning, we take human-provided annotations, which are grounded on intuition, to establish the “ground-truth”, and propose a method that can, without supervision, approximate this ground-truth, based on empirical measures. While obviously there still is a theoretical gap, and we are happy to discuss this and other theoretical limitations in the paper, it is still possible to make and measure progress, as we do here.
>
> > *There's no underlying reason why parts of an object that perform a distinct function should have uniform appearance that's different across different parts, and it's easy to think of examples where they don't: wheels of a car, wings of a butterfly, organs on a face (eyes, ears, nose, mouth), and so on.*
>
> Our main assumption is that similar parts  (e.g. the mouth for two people) are more similar _to each other_ than two different parts (e.g. an eye and a mouth). The uniform appearance prior is a very simple approximation of this idea and is complemented by the other priors (equivariance, distinctive appearance, correspondence) that we use.
>
> While this assumption is of course not universally applicable (i.e. for all parts of an object, for all objects), it is true often enough to be helpful: it leads to a considerable performance improvement over prior work (e.g. SCOPS [26]) that does not use this concept. We had experimented with more complex priors (e.g. higher level statistics or textures) but this did not result in meaningful improvements while adding considerable complexity.
>
> Optical flow, as used in [A], can also be a useful cue but is still imperfect. For instance, it would be very hard to identify the windshield of a car as a part because it would rarely if ever be observed moving relative to the rest of the car. In addition, measuring optical flow is a difficult problem in its own right, which in practice often relies on supervised learning. Finally, utilizing it for part segmentation is still very challenging due to camera motion, non-rigid deformations and other factors, which may explain why [A] is limited to relatively clean and visually simple data.
>
> In our case, our model can derive information from the _collection_ of images. Since our model learns to correlate parts across images, it is able to reliably identify parts even for instances that do not provide much signal for learning on their own (e.g. completely black birds such as ravens - see Fig.A.1). The reverse is also true, e.g. the wheels of a car are segmented as a single part despite their non-uniform appearance - see Fig. A.4.
>
> We believe that the choice of data (i.e. optical flow vs. image collections) already partially defines _what a part is_ as it is directly tied with the assumptions that can be made about the properties of the discoverable components. There is no right or wrong choice as both directions can learn to discover parts that align with human intuition but none can cover all imaginable aspects of the ambiguous nature of the general concept “part”.
>
> >  *the number of parts will be different for different categories, and may also vary within a category, yet here the number of parts (K) is a fixed element of the architecture. This means that an architecture will have to be tailored for a specific category, depending on what the authors think is the "true" number of distinct parts.*
>
> The question for “how to tune the number of parts/components” is fundamentally fused with unsupervised learning. Even in $K$-means, which is arguably the most commonly used grouping algorithm, $K$ needs to be hand-picked. Although several heuristics have been proposed in the last decades, usually they require some other hyper-parameter (e.g. expected density, number of neighbors, group size, etc.) that has a similar effect. This also touches the previous discussion of “what is a part” and “parts are ambiguous”. One can think of manually selecting $K$ as adding a prior on what _granularity_ of segmentation we are looking for. For quantitative evaluation and comparisons, we set $K$ to the same number of parts as previous methods (which is different from the ground truth). In the appendix we show results with different $K$ and find they are still meaningful. Additionally, our method works across different datasets (birds, horses, cars, humans, motorcycles, etc.) with virtually no changes.
>
>
> We hope that this explanation could shed light on our design choices and we will include parts of this discussion in relevant parts of the paper. We are glad that we can offer to respond to any additional questions in the newly introduced discussion period.
>
>
> -----
>
> [Gibson66] Gibson, J.J. and Carmichael, L. “The senses considered as perceptual systems” (Vol. 2, No. 1, pp. 44-73). Boston: Houghton Mifflin, 1966
>
> [MONet19] Burgess et al., “MONet: Unsupervised Scene Decomposition and Representation”, ArXiv 2019
>
> [SlotAttention20] Locatello et al., “Object-Centric Learning with Slot Attention”, NeurIPS 2020
>
> [GENESIS20] Engelcke et al., “GENESIS: Generative Scene Inference and Sampling with Object-Centric Latent Representations”, ICLR 2020
>
> [A]: Sabour, Sara, et al. "Unsupervised part representation by flow capsules." International Conference on Machine Learning. PMLR, 2021.
>
> [B]: Bear, Daniel M., et al. "Learning physical graph representations from visual scenes." arXiv preprint arXiv:2006.12373 (2020).
>
> [C]: Kipf, Thomas, Elise van der Pol, and Max Welling. "Contrastive learning of structured world models." arXiv preprint arXiv:1911.12247 (2019).
>
> [D]: Hénaff, Olivier J., et al. "Efficient visual pretraining with contrastive detection." arXiv preprint arXiv:2103.10957 (2021).

---

> > ### Comment · Reviewer_ogkV · 2021-08-25
> > **Thank you for the thoughtful response!**
> >
> > Thank you for your very well-thought out response to my concerns. It's clear that you've recognized many of the issues I raised and your choices for how to define parts and learn them were principled -- even though others (including me) might have intuitively favored different principles. Based on your discussion and your emphasis on improved performance on a standardized task, I am happy to raise my score. If this manuscript is accepted, I encourage you to include more of this discussion in the final version.
> >
> > _More detailed response_
> >
> > I agree that different methods may be needed to identify objects versus parts, depending on which definition of "part" you're using. The definition Gibson provides should work equally well for objects and for parts if the constraint is relaxed from "completely independent" to "partly independent" motion. For instance, I could imagine an unsupervised learning algorithm that identified objects based on motion completely independent of the rest of the scene, whereas parts only needed to be partly independent (e.g. an arm can move partly independently of a torso, but not completely so.) I accept that right now, there are probably no methods that can correctly decompose both a scene into objects and objects into parts without supervision -- but in principle, the use of motion/optical flow (as in [A] and [B]) seems promising to me.
> >
> > Your discussion of what makes a part a part is interesting and useful. I think it could play a larger role in the text itself. I also would appreciate seeing some examples of the more complex constraints besides "uniform color" to see that they don't lead to substantially better parts. As for using optical flow, I understand that there are technical challenges placing this beyond the scope of the present work (mainly because unsupervised optical flow prediction hasn't been solved in general.) For these reasons, it would help if you could lay out more clearly what assumptions you're making about what constitutes a part and how the dataset needs to be organized for your method to discover the parts.
> >
> > Finally, I agree that the choice about how to select K is not at settled matter. Maybe drawing more attention to the fact that you don't have to change K for different object classes, and showing that the method works for a decently large range of K values, would be sufficient for now.

---

### Official Review · Reviewer_DgiT · 2021-07-19

**Rating:** 7
**Confidence:** 4

**Summary:**

The authors present a new self-supervised framework for object part discovery, where two networks are jointly trained to represent and segment the image into parts, respectively. The networks are trained with a linear combination of several losses.

The segmentation network outputs a set of masks for each image, which correspond to the object parts in the image. The first term in the objective encourages features within a mask to be similar to each other.

Despite being unsupervised, these parts are consistent across images. The second term in the objective leverages this finding by constructing a contrastive objective which uses the same part (across images) as positive pairs, and different parts as negative samples.

Two other terms are added to the loss: equivariance to affine image transformations, and a "visual consistency" loss which penalizes the pixel-domain variance within masks.

The authors evaluate their method on several standard dataset, including CUB and DeepFashion.

**Limitations And Societal Impact:**

Yes, the authors adequately addressed the limitations and potential negative societal impact of their work.

**Main Review:**

Although somewhat involved technically and conceptually (the final loss contains 4 different terms), the method is well motivated and easy to follow. The ablations section shows that each term is indeed necessary and leads to large gains in performance. The comparison to prior art is very encouraging, with strong results presented throughout.

In addition to presenting an unsupervised method for part discovery, the authors investigate freeing themselves from supervised pretraining as well. There is indeed a large drop in performance, however this is an important baseline for future research in fully unsupervised part discovery.

One ablation is missing however: the authors define the positive pairs in their contrastive objective as the same part across images, rather than the same part across different views of the same image. How important is this modification? This finding could be useful beyond the specific problem considered here.

**Time Spent Reviewing:**

2

---

> ### Author Response · Authors · 2021-08-10
> **Response**
>
> We thank the reviewer for the encouraging feedback and the clear analysis of our method.
>
> > *One ablation is missing however: the authors define the positive pairs in their contrastive objective as the same part across images, rather than the same part across different views of the same image. How important is this modification? This finding could be useful beyond the specific problem considered here.*
>
> In traditional contrastive self-supervised learning the task definition is different; each image is an individual, distinct instance that needs to be contrasted to all other images. Here, we know that each image likely contains the same set of parts and we would like to find correspondences to obtain consistent labellings across images. Thus, while the formulation of the loss is the same, our task benefits from the ability to draw positives from _other_ images.
>
> We are running an experiment with a modified contrastive loss based on the reviewer’s suggestion. Instead of selecting positives from other images, we select positives from a different view of the same image, while leaving the negatives unchanged. The outcome of this experiment will show how important our formulation is to learn correspondences across images. Our preliminary results suggest that this performs worse than our model. We will post an update with the final results in the next few days.

---

> > ### Author Response · Authors · 2021-08-12
> > **Result update**
> >
> > The suggested ablation experiment on the influence on the contrastive loss with different views instead of different images on CUB dataset has now finished (please see Table 2 in the paper for comparison with the other baselines).
> >
> > |  Variant | FG-NMI  | FG-ARI  |
> > |---|---|---|
> > | no contrastive loss  | 45.2  | 20.5  |
> > | different views  | 47.6  | 21.3  |
> > | different images  | 49.1 | 21.9  |
> >
> > As expected the model can draw more information from a consistency loss across different images than from different views of the same image. However, using different views is still better than not using any contrastive loss at all. Thank you again for this interesting ablation idea. We will add the results and discussion to the paper.

---

> > > ### Comment · Reviewer_DgiT · 2021-08-26
> > > **Thank you for the additional ablation**
> > >
> > > Thank you for the additional ablation, it is very interesting that these cross-image positive pairs perform better than cross-view positive pairs. This is possible in a sense because of the restricted setting of the dataset (all images contain the same parts), but it isn't unreasonable to think it could be generalized to datasets used e.g. in semantic segmentation, where the same classes generally occur in all images, with some variation.
> > >
> > > This further strengths the paper, and I continue to recommend its acceptance.

---

### Official Review · Reviewer_U3ZE · 2021-07-19

**Rating:** 5
**Confidence:** 4

**Summary:**

This paper aims to segment parts of visual object classes in an unsupervised manner. The segmented part should be consistent within the class. To enable the model, several loss functions are added including different consistency losses, contrastive loss across different images, and equivariance loss on the transformation of the current image. Quantitative and qualitative experiments are conducted to examine the effectiveness of the proposed model.

**Limitations And Societal Impact:**

The paper has discussed limitations and social impart.

**Main Review:**

********************Strengths********************

The proposed approach for solving the part discovery is reasonable. More specifically, i) a consistency loss to encourage the uniformity within each part/mask; ii) a contrastive loss to encourage the consistency of parts across different images; iii) a loss to encourage the visual consistency within each part; iv) a loss to maintain consistency of features after transformation.

The code is provided, which can benefit the research along this direction.


********************Weaknesses********************

1.	It is unclear to me what is the potential applications of the studied topic and why it is important for studying. The proposed method relies on self-altered parameters including weights for loss and K for the number of parts, which is rather heuristic and hard to use in the real case. That is, the parameters need to be tuned each time when changing the dataset or number of parts.
2.	The writing is hard to understand in most of the model parts, especially for those descriptions of equations. For example, what does m mean in Eq (3), which often makes me confusing. Some model choice also needs more explanations, such as why use KL for transformation loss but not generally used L2. It is suggested to show some qualitative results on ablation studies to demonstrate the function of each loss.
3.	Why do the qualitative results only contain those compared with “SCOPS”, and how about the others listed in Table 1. Besides, the performance in Figure 4 is rather bad from my point of view, head of the dog, cat, and horse are even not in the same category of masks. This makes me wonder whether the problem has been solved properly.
4.	For the k-means baseline, why not try on some features learned by the general segmentation task to see how the model compared with them? Besides, it is suggested to put the ground truth used to calculate loss as an example for visualization, even this is an unsupervised method.


In summary, I am leaning towards rejection and do not think the paper meets the acceptance bar of NeurlPS. I suggest the author revise the paper to make it stronger.


**Time Spent Reviewing:**

10

---

> ### Author Response · Authors · 2021-08-10
> **Response**
>
> We thank the reviewer for the detailed feedback and suggestions. We address the raised points:
>
>
> > *It is unclear to me what is the potential applications of the studied topic and why it is important for studying.*
>
> The aim of our work is to obtain densely labelled parts of objects with minimal supervision. Part segments provide useful intermediate representations that are invariant to camera, lighting, object appearance and pose variation. Such representations are useful in analyzing objects in higher-level tasks, such as human-machine interaction, robotic manipulation etc. An example of immediate application of this is in 3D reconstruction [SS3D20] where SCOPS [26] is used to impose a semantic consistency constraint in 2D space. Our method can be used as a drop in replacement for [26]. Part-based representations have also been used extensively in fine-grained recognition, as we discuss in l.75-79. It is also of scientific interest to study how much unsupervised part discovery can align with human intuition. We will revise the introduction to and emphasize the importance of this research problem.
>
> > *The proposed method relies on self-altered parameters including weights for loss and K for the number of parts, which is rather heuristic and hard to use in the real case. That is, the parameters need to be tuned each time when changing the dataset or number of parts.*
>
> No. Our method is not as sensitive to hyper-parameter tuning on each dataset as the reviewer suggests. We point out (l.206-208) that we find the loss weights/hyperparameters on CUB, but use the same settings for all other datasets. The only exception is the equivariance term weight on PASCAL VOC. We found that increasing it improves performance, likely due to the fact that the training datasets for each individual PASCAL category are comparably small (~500 images) and this term acts similarly to data augmentation.
>
> $K$ can be interpreted as the ‘coarseness’ of the discovered parts and is inherently necessary in _unsupervised_ part segmentation. For example, a hand can be seen as one single part or composed of 6 parts: palm + 5 fingers. $K$ is a property of the object class and the desired granularity of the predictions and not the dataset. We will include these points in the discussion section.
>
>
> > *The writing is hard to understand in most of the model parts, especially for those descriptions of equations. For example, what does m mean in Eq (3), which often makes me confusing.*
>
> The $m$ in Eq. 3 is in fact $n$, thank you for pointing this out. In Eq (3) it is thus  $i \neq n$, which means all other images in the batch except the target image $n$. All other symbols have been formally introduced in the method section. We will add more examples and intuitions to the methods section to help understanding.
>
> > *Some model choice also needs more explanations, such as why use KL for transformation loss but not generally used L2.*
>
> We use KL divergence because the original and transformed outputs are pixelwise probability distributions over part classes. The KL divergence is a natural choice to compare two distributions. For example, [26] follows the same approach. To quantify, we have run experiments with L2 instead of the equivariance term. We find L2 loss performs slightly worse, reaching 46.5 NMI vs 49.1 NMI with KL divergence. For fair comparisons, we searched for the best loss coefficient of the L2 equivariance loss for this experiment.
>
> > *It is suggested to show some qualitative results on ablation studies to demonstrate the function of each loss.*
>
> Thank you for the suggestion, we will add a qualitative loss ablation figure to the appendix.
>
> > *Why do the qualitative results only contain those compared with “SCOPS”, and how about the others listed in Table 1.*
>
> Due to space constraints, we compared qualitatively to SCOPS [26] only, as it is the current state-of-the-art in unsupervised part segmentation (Huang and Li [25] use fine-grained image-level supervision). ULD is an unsupervised landmark discovery method and does not work well for segmentation, as shown in [26]. DFF is qualitatively similar to the k-means baseline, as it relies on feature clustering. We are happy to include a figure comparing all methods in the appendix.
>
> > *The performance in Figure 4 is rather bad from my point of view, head of the dog, cat, and horse are even not in the same category of masks. This makes me wonder whether the problem has been solved properly.*
>
> We point out that this is not a correct interpretation of Figure 4. As mentioned in Section 1 of the Appendix, we train separate models for _each_ of the 10 PASCAL categories (dog, cat, bicycle), so the outputs of different categories are not comparable. PASCAL is a challenging dataset due to the limited number of training samples, and at the same time, the high variation in object appearance. For all PASCAL categories our method is quantitatively better than the previous state of the art [26] (who also train one model per category). We note that it is not meaningful to train a single model for all categories as there is no reason why different object categories (e.g. cars and horses) should have common parts.
>
> > *For the k-means baseline, why not try on some features learned by the general segmentation task to see how the model compared with them?*
>
> The purpose of the $K$-means baseline is to show that simple clustering of ImageNet features leads to subpar part segments. Further motivation can be found in Sec. 3.2, where we qualitatively ablate the layer choice by comparing the quality of clustering VGG features.
>
> Following the reviewer’s, we also performed $K$-means clustering on features of a model trained for a general segmentation task. We used DeepLab-v2 (with ResNet-101 as backbone) trained for segmentation on PASCAL VOC. Clustering layer 5 features results in FG-ARI: 14.4 and FG-NMI: 30.0 for part segmentation on CUB, which is comparable to the baseline reported in our paper (Table 2, FG-ARI: 14.7 and FG-NMI: 34.9 ) that clusters features from an ImageNet-trained model, but significantly worse than our method (FG-ARI: 21.9, FG-NMI: 49.1).
>
> > *Besides, it is suggested to put the ground truth used to calculate loss as an example for visualization, even this is an unsupervised method.*
>
> We did so because of space constraints. We thank the reviewer for this suggestion, we will make this change for the images in the appendix.
>
> ---
>
> [SS3D20] Li et al., “Self-supervised Single-view 3D Reconstruction via Semantic Consistency”, ECCV 2020

---

> > ### Comment · Reviewer_U3ZE · 2021-08-23
> > **Thanks for the response**
> >
> > Thank you so much for the response. The response has addressed some of my concerns, and thus I would like to raise my score to reflect this. However, I am still leaning towards rejection. The hyperparameter setting (e.g. K) and results (e.g. Fig.4) still make me doubt the flexibility and usability of the proposed methods. I sincerely hope authors can revise the paper based on the review and rebuttal to make the submission stronger.

---

> > > ### Author Response · Authors · 2021-09-01
> > > **Thank you!**
> > >
> > > We thank the reviewer for raising their rating from 4 to 5 after our response. We would like to point out our response to reviewer ogkV with respect to selecting K. It is a natural hyper-parameter that is inherent to all unsupervised grouping algorithms (e.g. k-means). It controls the granularity of the prediction and there is no right or wrong choice. It depends on the user to specify the desired coarseness of the subdivision.
> > >
> > > **Results in Fig.4:**
> > >
> > > Like all unsupervised methods there is still room for improvement. However, as we and the other reviewers point out, the presented method shows a large improvement over all previous methods in the quantitative experiments. We thus deem the approach of interest to the community to advance the state of the art at this very challenging task.
> > >
> > > > *I sincerely hope authors can revise the paper based on the review and rebuttal to make the submission stronger.*
> > >
> > > We will include all the feedback and additional results from the review and discussion period in the final version of the paper. The code and trained models will also be available for full reproducibility.

---

### Decision · Program_Chairs · 2021-09-27

**Decision:**

Accept (Poster)

**Comment:**

This submission initially received mixed reviews (two weak rejects, one weak accept). After the rebuttal, the reviews remain mixed (two weak accepts, one weak reject).  The AC has carefully read the paper, reviews, and rebuttals.  The AC agrees with the concerns raised by the reviewers, including the definition of parts, the lack of discussion on methods using motion, and the missing discussion on the number of parts (K). The AC also believes this submission has made enough contributions to warrant acceptance, given its technical innovations, results, and analyses. In the camera ready, the authors should definitely discuss the related work on using motion to discover object parts, in particular those raised by reviewer ogkV:

[A]: Sabour, Sara, et al. "Unsupervised part representation by flow capsules." International Conference on Machine Learning. PMLR, 2021.

[B]: Bear, Daniel M., et al. "Learning physical graph representations from visual scenes." arXiv preprint arXiv:2006.12373 (2020).